# Antibiogram Profile of *Acinetobacter*
*baumannii* Recovered from Selected Freshwater Resources in the Eastern Cape Province, South Africa

**DOI:** 10.3390/pathogens10091110

**Published:** 2021-08-31

**Authors:** Mary Ayobami Adewoyin, Kingsley Ehi Ebomah, Anthony Ifeanyi Okoh

**Affiliations:** 1SAMRC Microbial Water Quality Monitoring Centre, University of Fort Hare, Alice 5700, South Africa; kebomah@ufh.ac.za (K.E.E.); AOkoh@ufh.ac.za (A.I.O.); 2Applied and Environmental Microbiology Research Group, Department of Biochemistry and Microbiology, University of Fort Hare, Alice 5700, South Africa; 3Department of Environmental Health Sciences, College of Health Sciences, University of Sharjah, Sharjah P.O. Box 27272, United Arab Emirates

**Keywords:** *A. baumannii*, multiple drug resistance, antibiotic-resistant genes, freshwater

## Abstract

*Acinetobacter* species have been found in a variety of environments, including soil, food, plants, hospital environments and water. *Acinetobacter baumannii* is an opportunistic and emerging waterborne pathogen. It has been implicated in several nosocomial infections that demonstrate resistance to commonly administered antibiotics. We investigated phenotypic antibiotic resistance (PAR) and relevant antibiotic resistance genes (ARGs) in *A. baumannii* isolated from three freshwater resources in the Eastern Cape Province, South Africa; *A. baumannii* (410) was confirmed by the *recA* and *gyrB* genes of 844 suspected *Acinetobacter* species in the water samples. The PAR of the confirmed isolates was assessed using a panel of 11 antibiotics by the disc diffusion method, while ARGs were investigated in isolates exhibiting PAR. The *A. baumannii* isolates were resistant to piperacillin-tazobactam (11.2%), ceftazidime (12%), cefotaxime (18.8%), cefepime (8.8%), imipenem (2.7%), meropenem (4.15%), amikacin (2.4%), gentamicin (8.8%), tetracycline (16.8%), ciprofloxacin (11%) and trimethoprim/sulfamethoxazole (20.5%). For multidrug resistance (MDR), two isolates were resistant to all antibiotics and 28 isolates were resistant to imipenem and meropenem. Moreover, β-lactamases *bla*_TEM_ (64.4%) and *bla*_OXA-51_ (28.70%) as well as sulphonamides *sul1* (37.1%) and *sul2* (49.4%) were common ARGs. Overall, PAR and ARGs had positive correlations (*r*) in all rivers. Detection of MDR-*A. baumannii* in freshwater resources could be linked to possible wastewater discharge from the nearby animal farms, indicating potential implications for public health.

## 1. Introduction

*Acinetobacter* species are known lactose non-fermenters that were formerly regarded as colonisers of different natural habitats, with low ability to cause infection [1,2]. Recent reports have shown *A. baumannii* is the most virulent species in the genus *Acinetobacter*, causing severe infections among patients on prolonged admission in hospital intensive care units (ICUs) [3]. Similarly, individuals with diabetes and compromised immune systems are prone to *A. baumannii* infections [4,5,6]. Some of the reasons for the rise in *Acinetobacter* species infections are their ability to adapt to harsh environmental conditions (e.g., dry surfaces) and their resistance to different commonly prescribed antimicrobial agents [7,8].

Antibiotic resistance in bacterial strains is a global health problem of major concern. The cause for concern with regard to antimicrobial resistance is the failure of some front-line and last-line antimicrobial agents to combat the rising resistance in pathogenic microorganisms, including *A. baumannii* [7,8,9]. In addition, incidences of multidrug-resistant *A. baumannii* (MDRAB) infections are now happening on a global scale [10]. The mechanism of antibiotic resistance in pathogenic bacteria could be intrinsic or acquired through horizontal gene transfer (HGT) or genetic mutations [11]. *A. baumannii* possesses in its chromosome genes the code for enzymes and efflux pumps that is responsible for its high natural resistance [12,13]. The tendency for *A. baumannii* and other related species to acquire resistance-encoding genes [14] from other closely related species even of different genus increases their resistance profile. These genes are often the basis for the acquisition of new antimicrobial resistance genes (ARGs), as they themselves or their regulators are subject to mutations. 

Therefore, ARGs are essential determinants for pathogenic *A. baumannii* to thrive in the host system. For example, resistance to β-lactams was associated with the expression of beta-lactamase genes such as *bla*_TEM_, *bla*_SHV_, *bla*_OXA-51_, *bla*_CTX-M (GROUP 1)_ and *bla*_CTX-M (GROUP 2)_ and a few other ARGs harboured by *Acinetobacter* spp. [15,16,17]. Similarly, the genes responsible for antibiotic resistance against aminoglycosides (*aacC2, aphA1* and *aphA2*), fluoroquinolones (*qnr*A, *qnr*B, *qnr*D and *qnr*S), sulfonamides (*sul1* and *sul2*) and tetracyclines (*tet*(A), *tet*(B), *tet*(O), *tet*(L) and *tet*(M)) were previously reported in *A. baumannii* [18].

This study investigated the antibiogram profile as well as antibiotic resistance genes (ARGs) of *A. baumannii* species recovered from Great Fish, Keiskamma and Tyhume Rivers in the Eastern Cape Province, South Africa as shown in Figure 1. The isolation of pathogenic *A. baumannii* strains from freshwater sources have been reported [19,20,21,22,23]. The goal of this study was to evaluate the impact of the anthropogenic activities such as in-flow of wastewater treatment plant (WWTP) effluent, animal husbandry and indiscriminate disposal of waste to the freshwater resources in that area. These human activities have the tendency to expose existing bacterial species to antibiotics and influence the behaviour of the microbial communities in these rivers. 

## 2. Results

### 2.1. Identification and Antibiotic Resistance Profile of Acinetobacter Species

Eight hundred and forty-four (844) presumptive isolates of *Acinetobacter* species were selected for molecular identification and confirmation. The results in Appendix A show that 410 (48.58%) isolates were confirmed to be *A. baumannii* across the three rivers, from which 153, 102 and 155 isolates were recovered from Great Fish, Keiskamma and Tyhume Rivers, respectively.

### 2.2. Phenotypic Antibiotic Resistance (PAR) Profile of A. baumannii

While the majority of the confirmed *A. baumannii* isolates were susceptible to the test antibiotics (Appendix A), a certain percentage of the isolates were resistant to the antimicrobial agents (Table 1). According to the values obtained in comparison to CLSI guidelines for antimicrobial susceptibility testing for *A. baumannii*, the isolates were classified as susceptible (S), intermediate resistant (I) or resistant (R). However, to better explain the outcome of the antibiotic susceptibility testing, a non-susceptible (R + I) column was created in this study, which gives a summarized view of the level of resistance to each of the antibiotics used. Although resistance to SXT 84 (20.5%) was higher than other antibiotics, there was no significant difference (*p* < 0.05) when compared to other antimicrobial agents. Nonetheless, non-susceptibility (R + I) or intermediate resistance (I) to CTX was significantly higher (*p* < 0.05) than other antibiotics, followed by CAZ, which also belongs to the same class of antimicrobial agent-cephems. Comparably, CTX and TET had a similar level of resistance. Notably, at least ten *A. baumannii* isolates were resistant to each of the antibiotics tested, while 11 (2.7%) and 17 (4.15%) of the *A. baumannii* isolates were resistant to imipenem and meropenem, respectively.

### 2.3. Multiple Drug Resistance (MDR) Phenotypes

The multiple drug resistance (MDR) phenotypes of *A. baumannii* isolated from the freshwater sources (Great Fish, Keiskamma and Tyhume Rivers) are highlighted in Figure 1 (heatmap). *A. baumannii* exhibited some degree of MDR phenotypes, ranging from three to seven of the antimicrobials used. However, out of 410 *A. baumannii* isolates, 54 (13.17%) exhibited multiple drug resistances. Figure 1 describes the patterns of MDR observed in this study according to source of isolation of the *A. baumannii*. Of the 54 MDR *A. baumannii* isolates, 16 (29.63%) were resistant to three classes of antibiotics, 12 (22.22%) were resistant to four, and 11 (20.37%) were resistant to five and six, while four (7.41%) were resistant to all (seven) of the classes of the antimicrobial agents. The highest MDR value was observed in sampling site KE2 (19; 35.19%) followed by site GF5 (10; 18.52%) and site GF4 (6; 11.11%). There were no MDR phenotypes observed in the remaining sampling sites.

### 2.4. Profile of ARGs among A. baumannii

Figure 1 illustrates the collection of water sample from the three rivers, the antibiotic susceptibility testing and resistance genes in *A. baumannii* isolates. Table 2 summarizes the phenotypic antibiotic resistance (PAR) and ARGs of the *A. baumannii* recovered in this study. Both *sul1* and *sul2* genes were investigated in 83 sulfonamide-resistant *A. baumannii*. Appendix A shows the gel picture of some *A. baumannii* isolates harbouring *sul1* and *sul2* resistance genes. Among the screened isolates, 32 (37.1%) possessed *sul1*, 43 (49.4%) harboured *sul2* (Table 2), 18 (21.7%) had both genes, while 26 (31.3%) isolates did not harbour any of the genes. Figure 2 (heatmap below) outlines the distribution patterns of the ARGs among the isolates recovered from the river water samples, and this shows that the Great Fish River possessed the highest number of *A. baumannii* isolates that harboured ARGs. However, factorial analysis of variance (ANOVA) indicated no significant difference (*p* < 0.05) between *sul1* and *sul2* ARGs in the Great Fish and Tyhume Rivers, whereas the presence of *sul2* in *A. baumannii* was significantly higher (*p* < 0.05) than *sul1* in the Keiskamma River water samples.

Similarly, Appendix A shows the gel picture of some *A. baumannii* isolates harbouring *apHA1* and *apHA2* resistance genes. Among 40 *A. baumannii* isolates evaluated for aminoglycoside-resistance genes (*apHA1* and *apHA2*), 13 (25.5%) and 21 (41.2%) possessed *apHA1* and *apHA2* genes, respectively (Table 2), two isolates possessed all the genes, and 15 did not harbour any genes. From ANOVA, detection of *apHA1* gene in the Great Fish River and *apHA2* gene in the Keiskamma River showed higher significance (*p* < 0.05) when compared to other aminoglycoside resistance genes.

Five relevant fluoroquinolone resistance genes, *qnrA*, *qnrB*, *qnrC*, *qnrD* and *qnrS*, were investigated in the *Acinetobacter* spp. (Appendix A and Table 2). Forty-four *A. baumannii* isolates were screened for the presence of the selected ARGs, and only *qnrB* and *qnrD* were detected in all the isolates. Out of 44 isolates, only seven (15.9%) harboured *qnrB* and eight (18.2%) harboured *qnrD* genes, three (6.8%) possessed both *qnrB* and *qnrD* genes, and 35 (79.5%) did not possess either of the two genes. Isolates from both the Great Fish and Tyhume River water samples harboured *qnrD*, while some isolates from all the water samples possessed *qnrB* (Figure 2). However, while no significant difference (*p* < 0.05) was seen among the *A. baumannii* isolates that harboured *qnrB* in the samples, those that possessed *qnrD* genes in the Great Fish (*p* < 0.05) and Tyhume (*p* < 0.05) Rivers were significantly higher (*p* < 0.05) than other fluoroquinolone resistance genes investigated in all the rivers. 

Seventy isolates of *A. baumannii* that exhibited phenotypic resistance to tetracyclines were screened for six tetracycline resistance genes (Appendix A and Appendix A). Among the *tet* genes, *tet*(B) was more prevalent and was detected in 19 (27.1%) isolates, followed by *tet*(A) in eight (11.4%) isolates; other tetracycline resistance genes were in the following order: *tet*(L), three (4.3%) isolates; *tet*(C), 2 (2.9%) isolates; *tet*(M), two (2.9%) isolates (Table 2). None of the isolates harboured *tet*(O), while 46 (63%) isolates did not harbour any of the *tet* genes. Similarly, *tet*(B) was significantly higher (*p* < 0.05) than all other tetracycline resistance genes in both the Great Fish and Keiskamma Rivers, whereas a significant number of *A. baumannii* isolates bearing the gene were recovered from the Keiskamma River compared to the Great Fish and Tyhume Rivers. 

The β-lactamase resistance genes were harboured by 101 *A. baumannii* isolates. These isolates demonstrated phenotypic resistance to β-lactam antibiotics, including carbapenems. β-lactamase resistance genes, namely *bla*_TEM_*, bla*_SHV,_
*bla*_CTX-M_ (*bla*_CTX-M-1group_, _2group_, *bla*_CTX-M-9group_ and *bla*_CTX-M-8/-25group_), *bla*_VEB_*, bla*_GES_*, bla*_PER,_
*bla*_OXA-48-like_, *bla*_OXA-51_, *bla*_VIM_, *bla*_IMP_ and *bla*_KPC_**,** were investigated, and the results are highlighted in Figure 2 and summarized in Table 2. Appendix A show the gel pictures of some *A. baumannii* isolates harbouring β-lactamase resistance genes. All the ARGs except *bla*_OXA-23_, *bla*_OXA-40_, *bla*_OXA-58_, *bla*_CTX-M-9group_ and *bla*_IMP_ were detected in at least one isolate of *A. baumannii* recovered from the river water samples. However, *bla*_TEM_ was prevalent (65; 64.4%) followed by *bla*_OXA-51_ (29; 28.71%) among the *A. baumannii* isolates, while 12 (11.9%), 13 (12.9%), 12 (11.9%), 12 (11.9%), two (2.0%), 10 (9.4%), nine (8.9%), five (4.9%), and three (2.9%) harboured *bla*_CTX-M-8/-25_, *bla*_GES_, *bla*_SHV_, *bla*_CTX-M-1group_, *bla*_CTX-M-2group_, *bla*_OXA-48-like_, *bla*_KPC_, *bla*_VEB_, and *bla*_PER_, respectively. While *bla*_VEB_*, bla*_IMP_, *bla*_CTX-M-8/-25group_, and *bla*_OXA-48-like_ were predominantly recovered from the Tyhume River water samples, *bla*_SHV_, *bla*_GES_, *bla*_KPC_ and *bla*_CTX-M-1group_ were detected more in the Great Fish and Keiskamma River water samples (Figure 2).

β-lactamase resistance genes indicated that the recovery of the *bla*_TEM_ gene was significantly higher (*p* < 0.05) than other β-lactamase genes investigated in this study, using Tukey HSD (preferred post hoc). Additionally, isolates harbouring *bla*_TEM_ in the Great Fish River were significantly higher than those recovered from the Keiskamma (*p* < 0.05) or Tyhume (*p* < 0.05) Rivers. However, *bla*_SHV_, *bla*_CTX-M-1group_, *bla*_CTX-M-8/-25group_ and *bla*_GES_ harboured by the isolates from the Great Fish River were significantly higher (*p* < 0.05) than those from the Keiskamma or Tyhume Rivers. Similarly, there was no significant difference (*p* < 0.05) between *A. baumannii* that harboured *bla*_OXA-51_ in the Great Fish and Keiskamma Rivers, whereas isolates bearing *bla*_OXA-51_ in both rivers were significantly higher (*p* < 0.05) than isolates from the Tyhume River.

Eleven *A. baumannii* were resistant to imipenem, while 17 were resistant to meropenem, which is usually the last choice of antimicrobial agents to treat infections caused by *Acinetobacter baumannii*. Some of the isolates that were resistant to both meropenem and imipenem harboured *bla*_TEM_, *bla*_SHV_, *bla*_CTX-M-1group_, *bla*_OXA-48-like_ and *bla*_OXA-51_ genes. 

### 2.5. Correlation between PAR and ARGs

Correlation in this study was indicative of the strength of the relationship between the two variables—PAR and ARGs—in which ARGs were assumed to be the likely cause or contributed to the PAR observed in *A*. *baumannii* in each of the rivers. The summary of PAR and ARGs in *A. baumannii* recovered from specific sampling sites in the rivers studied is presented in Table 2, while Figure 3 describes the correlation between the PAR and ARGs of *A. baumannii* across the rivers studied. The linear relationship was determined using Pearson’s correlation coefficient (*r*). The coefficients of correlation (*r*) between the PAR and ARGs were generally positive in all the rivers, where 0.0456, 0.2131 and 0.0206 were determined for Great Fish, Keiskamma and Tyhume Rivers, respectively. However, the *p*-values of the correlation between PAR and ARGs in the Great Fish and Tyhume Rivers were 0.6643 and 0.8397, respectively, indicating weak correlations. The *p*-value in the Keiskamma River was 0.0403, which suggested a significant (*p* ≤ 0.05) relationship. 

## 3. Discussion

The antibiogram profile of *A. baumannii* evaluated in this work revealed MDR isolates at several sites in three rivers in the Eastern Cape of South Africa (Figure 1), which might be due to the anthropogenic impact in the study area. While *A. baumannii* in the freshwater resources were generally susceptible to antibiotics, few of the isolates exhibited MDR, and the correlation between the PAR and ARGs was positive in all the rivers. Fernando et al. [24] and Kittinger et al. [21] reported high susceptibility to the antimicrobial agents by *A. baumannii* from the natural environment, whereas Tsai et al. [25] emphasized resistance to antibiotics (SXT) belonging to the class of folate pathway inhibitors in *Acinetobacter* species recovered from water resources. This further validated the claim that non-clinical *A. baumannii* are more susceptible to antimicrobial agents than clinical isolates [26,27].

In addition, a few *A. baumannii* isolates demonstrated MDR, exhibiting resistance to meropenem and imipenem similar to a study conducted by [1]. In that study, some isolates (*A. baumannii*) were 100% resistant to all tested antibiotics, including meropenem and imipenem. As such, our findings, in accordance with their surveys, suggested an additional risk to the global therapeutic battle. Nonetheless, this may represent an index of secondary exposure of water sources to drugs; alternatively, these isolates originated from sources where the use of antibiotics is grossly abused [28,29]. Summarily, MDR phenotypes revealed that Keiskamma sampling point 2 (KE2) was a hotspot of antibiotic-resistant *A. baumannii* in this study.

For the ARGs studied, β-lactamases (*bla*_TEM_) were the most prevalent genes harboured by *A. baumannii*, as was previously shown by [30,31,32]. Apparently, some of the *A. baumannii* strains that were resistant to carbapenems (IMI and MEM) possessed *bla*_TEM_, *bla*_SHV_, *bla*_VEB_, *bla*_SHV_, *bla*_CTX-M-1_, *bla*_OXA-48-like_ and *bla*_OXA-51_ (Table 2). As noted above, Maravic et al. showed that carbepenem-resistant strains of *Acinetobacter* species isolated from freshwater bore *bla*_TEM_ [1]. Therefore, apart from other reasons, possession of these genes could induce resistance to carbapenems (IMI and MEM), β-lactams (PTZ), and cephems (CAZ, CTX and CPM) in *A. baumannii* because these isolates typically exhibited MDR to all the four Ambler classes (A, B, C and D) of β-lactamases [33,34].

Additionally, *sul1* and *sul2* were common in the *A. baumannii.* For instance, the resistance to SXT that carried the highest percentage [25,31] correlated to the prevalence of sulfonamide resistance genes, particularly in sites KE2, GF4 and GF5 (Table 2). Compared to this study, the prevalence of sulfonamide resistance was reported in *A. baumannii* [35,36] while [37] showed that *A. baumannii* uses plasmid as a medium of transfer/acquisition of *sul2* resistance genes. 

Likewise, the detection of *tet*(B) in *A. baumannii* isolates among other tetracycline resistance genes was prominent. Among the sites investigated, KE2 had the highest number of *A. baumannii* strains that possessed this gene. The efflux system or protection of the ribosomal system has been the channel of resistance to tetracycline by *A. baumannii* [25]. The gene coding for both *tet*(A) and *tet*(B) is involved in the tetracycline efflux system and has been detected previously in both aquatic and clinical *A. baumannii* isolates [38,39,40,41]. In addition to protection of the ribosome by *tet*(M) [17,36,42], the gene is also used to mediate resistance to tetracycline, but was found in only two isolates in this study. 

In the same way, both *qnrB* and *qnrD* were detected in *A. baumannii*, though *qnrD* was more prevalent than other fluoroquinolone genes. Some of the *A. baumannii* isolates bearing the *qnrB* and *qnrD* genes also showed PAR to ciprofloxacin, especially isolates that were recovered from GF5 (Table 2). The ARGs could mediate resistance to CIP by inhibiting the activity of bacterial-DNA gyrase (encoded by *gyrA* and *gyrB* genes) and topoisomerase IV (encoded by *parA* and *parC* genes) [38,43,44,45].

Considering the presence of aminoglycoside resistance genes in *A. baumannii*, while *apHA1* and *apHA2* were recovered from GF4 at a lower amount compared to the amount at KE2 (Table 2), PAR to amikacin and gentamicin was the least among the antibiotics used (Table 1). All the same, a few A. *baumannii* isolates harboured the two aminoglycoside resistance genes, suggesting the possibility of HGT between different bacterial species. Nonetheless, the inactivation of the antibiotics by certain aminoglycoside-modifying enzymes such as phosphotransferases, acetyltransferases and adenyltransferases [38,45,46,47] have been linked to resistance. These modifying enzymes are produced by plasmids, integrons or transposons [35], which could be acquired via HGT [27,28]. 

Generally, *bla*_TEM_ was the most prevalent ARG detected in *Acinetobacter* spp. in all the three rivers studied, with its highest recovery from sampling site GF5, followed by the *sul*2 gene, which had its highest detection at sampling site KE2. In addition to the MDR phenotypes and ARGs, there was a positive correlation between PAR and ARGs in *A. baumannii* across the rivers.

Overall, the differences observed in the recovery of isolates exhibiting phenotypic resistance as well as resistance genes in one river compared to another could be a function of individual exposure, intrinsic factors such as gene transfer, porin loss and upregulations of efflux pumps, or human interactions and accumulation of anthropogenic activities at a particular sampling site. Future studies on the rivers should be conducted to examine residual antibiotics in the freshwater resources.

## 4. Materials and Methods

### 4.1. Description of Sampling Areas

Water samples were collected from three rivers, namely the Great Fish, Keiskamma and Tyhume Rivers, located at Chris Hani and Amathole District Municipalities of the Eastern Cape Province, South Africa, between April 2017 and March 2018. Five sampling points along each river course were selected for the study, based on different visible activities such as in-flow of wastewater plant (WWTP) final effluents, irrigation farming, fishing and animal rearing in the communities. 

Figure 4A shows the coordinates of the Great Fish River in the Chris Hani District Municipality of the Eastern Cape Province and the five sampling points. At GF1 there is irrigation farming and fishing, whereas GF2 is characterized by fishing, crop farming and animal rearing. Similarly, site GF3 is used for swimming, animal rearing and domestic purposes, while GF4 is a downstream of Craddock WWTP, where animal rearing and refuse dumping occurs. GF5 is known for fishing, recreational activities such as swimming, traditional ritual washing and dumping of refuse. The Great Fish River is major River that is used for irrigation and livestock farming in the area. The study sites are characterized by fish farming, irrigation, animal rearing, refuse dumping as well as effluents from WWTPs in Craddock. Figure 4B,C describe the flow paths of the Keiskamma and Tyhume Rivers. Both rivers are located in the Amathole District Municipality of the Eastern Cape Province and are exposed to different anthropogenic activities. For instance, crop farming and animal rearing are the normal activities at KE1, while KE2 is used for domestic activities and is also affected by wastewater pipe leakage and animal rearing. KE3 receives community runoff, wastewater pipe leakage, and domestic refuse. In addition, KE4 and KE5 are located downstream of Sandile Dam and a WWTP; animal husbandry and crop farming also take place at these locations. In the Tyhume River, sampling site TY1 is a recreational site where tourists visit and swim. Site TY2 is used for domestic activities, animal rearing and other farming activities. Both TY3 and TY4 are used for fishing, recreational activities and farming, whereas TY5 is located downstream of hospital waste discharge and waste discharge from the University of Fort Hare. 

The sampling sites and the corresponding sampling points are highlighted in the maps in Figure 4A–C. Water samples were collected aseptically in sterile 1 L glass bottles from the different sampling points along the river courses by midstream dipping of sample bottles at 25–30 cm down the water column, with the mouth tilting against the flow of the river. All the samples were labelled appropriately and transported to the Applied and Environmental Microbiology Research Group (AEMREG) laboratory in an ice chest and processed within six hours of collection.

### 4.2. Isolation and Molecular Confirmation of A. baumannii

Water samples were collected from Great Fish, Keiskamma and Tyhume Rivers in Eastern Cape Province of South Africa between April 2017 and March 2018 (Figure 4A–C). Isolation of the presumptive *Acinetobacter* species was carried out according to American Public Health Association [48] usingselective medium (CHROMagar *Acinetobacter* base) containing selective supplements (CHROMagar, Paris, France) at 37 °C for 24 h. The selective medium was prepared as instructed by the manufacturer. After incubation, 20 distinct *A. baumannii* colonies per sampling site were subcultured on nutrient agar (Oxoid, Hampshire, UK) using a streak plate method and then incubated at 37 °C for 24 h. Fifty percent (50%) glycerol stocks of the pure culture were prepared and stored at −80 °C. DNA extraction from the bacterial isolates was carried out using the direct boiling method. Molecular identification of Genus *Acinetobacter* was performed by targeting *Acinetobacter recA* gene using a forward primer, P-rA1 (5′-CCTGAATCTTCTGGTAAAAC-3′), and a reverse primer, P-rA2 (5′-GTTTCTGGGCTGCCAAACATTAC-3′), while for the confirmation of *A. baumannii,* the primer sets P-*Ab-ITS* (5′-CATTATCACGGTAATTAGTG-3′–forward; 5′-AGAGCACTGTGCACTTAAG-3′-reverse) were used according to [49], while reference strain DSM-102929 was used as the positive control. The components of the PCR assay are as follows: total volume of 25 µL reaction mixture contained 12.5 µL Taq PCR Master Mix (Qiagen, Hilden, Germany), 1.0 µL each of the 100 µM primers (Inqaba Biotech., Pretoria, South Africa), 6.5 µL nuclease-free water and 5.0 µL DNA templates. The condition for the PCR amplification included an initial denaturation step (94 °C, 5 min), followed by 35 cycles (92 °C, 40 s), annealing (58 °C, 40 s), and the final extension step (72 °C, 10 min) was performed using a thermocycler (Bio-Rad Thermal cycler, Hercules, CA, USA). Five microlitres (5 µL) of the amplicon was subjected to gel (1.5% agarose) electrophoresis at 100 Volts for 45 min in Tris Boric EDTA buffer (pH 8.0) (0.089 M Tris, 0.089 M boric acid, and 0.002 M EDTA). Ethidium bromide (5 µL of 0.5 mg/mL) (Sigma-Aldrich, St. Louis, MO, USA) was used for gel staining, and DNA ladder (100 bp) (Thermo Scientific, Vilnius, Lithuania) was added into the gels as a standard. Finally, DNA bands were visualized under an ultraviolet transilluminator (Alliance 4.7, Cambridge, UK).

### 4.3. Antibiotics Susceptibility Testing (AST)

Susceptibility of *A. baumannii* isolates to the commonly prescribed antimicrobial agents (Table 1) was determined using the disc diffusion method [50]. After 18 h of culture on nutrient agar (NA) plates, a few distinct colonies were picked and transferred into a test tube containing 10 mL of sterile normal saline (0.85%) and then vortexed gently to allow a proportional assortment of the bacterial cells. The turbidity of the bacterial suspensions was adjusted to 0.5 McFarland standards. A sterile swab stick was dipped into bacterial suspension and used to inoculate the freshly prepared Mueller Hinton agar (MHA) plate by spreading evenly on the agar surface. After air-drying the inoculated plates, the antibiotic discs were impregnated onto them using a sterile disc dispenser and incubated at 37 °C for 20 to 24 h [50]. The diameter of the zone of antibiotic inhibition was rounded to the nearest millimetre, while the isolates were classified as susceptible (S), intermediate resistant (I) or resistant (R) according to the values obtained in comparison to CLSI guidelines for antimicrobial susceptibility testing for *A. baumannii*. A panel of eleven antibiotics was selected from different antimicrobial groups or classes (Table 3) as representative drugs for the treatment of *A. baumannii* infections. Multidrug resistance (MDR) phenotype is defined as the resistance of isolates to at least three classes of antimicrobial agents (Table 3).

### 4.4. Screening of A. baumannii for Antibiotic Resistance Genes (ARGs)

*A. baumannii* that exhibited phenotypic resistance to test antibiotics were screened for antibiotic resistance genes (ARGs) using multiplex or singleplex PCR assays. The list of primers used for the detection of different ARGs and PCR conditions are presented in Appendix A. An aliquot of 25 μL reaction mixture containing a Master Mix (12.5 μL) (Thermo Scientific, Vilnius, Lithuania), oligonucleotide primer (0.5 μL each) (Inqaba Biotech, Pretoria, South Africa), nuclease-free water (6.5 μL) and template DNA (5 μL) was constituted for singleplex PCR assay. The multiplex PCR of 50 μL reaction volume was comprised of Master Mix (25 μL) (Thermo Scientific, (EU) Lithuania), each oligonucleotide primer (1 μL each) (Inqaba Biotech, Pretoria, South Africa), nuclease-free water (14 μL) and template DNA (5 μL). A thermocycler (Bio-Rad Thermal cycler, Hercules, CA, USA) was used for all PCR analysis. A reference strain of *A. baumannii* (DSM 102929) was used as positive control throughout the ARG screening. Ethidium bromide (5 µL of 0.5 mg/mL) (Sigma-Aldrich, St. Louis, MO, USA) was used for gel staining, and 5 µL of DNA ladder (100 bp) (Thermo Scientific, Vilnius, Lithuania) was added as a standard. Five microlitres (5 µL) of the amplicons were dispensed into the wells of the gel (1.5% agarose) and placed in Tris Boric EDTA buffer (pH 8.0) (0.089 M Tris, 0.089 M boric acid, and 0.002 M EDTA). The condition used for electrophoresis was 100 Volts for 45 min. Finally, DNA bands were viewed under an ultraviolet transilluminator (Alliance 4.7, Cambridge, UK).

### 4.5. Statistical Analysis

Factorial analysis of variance (ANOVA) was done to determine the variations in the ARGs harboured by *Acinetobacter* isolates across the three rivers studied in Statistica 13 software, using the Tukey HSD post hoc test (*p* < 0.05) to identify datasets that are significantly different from their counterparts. Correlations (Pearson) and test of significance were considered statistically significant when *p* values were ≥95% using Statistica v13 software. Pearson correlation coefficient was used to determine the strength of the linear association between the two variables—PAR and ARGs—in the rivers. The heatmaps were created in Microsoft Excel v16 to highlight the value of MDR phenotypes and ARGs.

## 5. Conclusions

The detection of ARGs in *A. baumannii* recovered from the aquatic environments in the study area might be an indication of the secondary impact of antibiotic mismanagement, which presents a significant challenge to public health. This could be the case, especially at sampling site GF4, which is characterized with discharge of wastewater effluent from the downstream Craddock WWTP, animal rearing and refuse dumping. The GF5 site is also exposed to community runoff and a dumpsite where domestic refuse is dumped at the river bank for biodegradation. KE2 is subject to domestic wastewater sludge, wastewater pipe leakage and animal rearing. Antibiotics such as tetracycline, penicillin, noritrine and the sulfonamide group are commonly administered to livestock in the area.

In this study, MDR was observed, and a positive correlation was detected between the PAR and ARGs of the *A. baumannii* strains recovered from the freshwater resources; in the case of the Keiskamma River, this was significant. The significant relationship between PAR and ARGs in the Keiskamma River could be an indication of selective pressure on antibiotics in that area. It was also evident that some of the isolates were resistant to ciprofloxacin, carbapenems and few other antibiotics. The prevalence of genes encoding for resistance to β-lactamase (*bla*_TEM_ and *bla*_OXA-51_), sulphonamide (*sul1* and *sul2*) and other antimicrobials might be responsible for the resistance as well as exposure to antibiotics due to anthropogenic activities, especially overuse of drugs in the livestock farming in the sampling areas. However, some of the resistant strains might be using other mechanisms to mediate resistance to antimicrobial agents in addition to harbouring the resistance genes. As such, the sampling sites where most of the ARG-bearing isolates were recovered could be categorised as hotspots for the pathogens. Thus, necessary measures should be taken against indiscriminate disposal of waste into the environment to prevent exposure of waterborne pathogens to unused or expired antibiotics and the propagation of resistant pathogens into the water sources.

## Data Availability

The data presented in this study are available on request from the corresponding author. The data are not publicly available because the authors still need the data to further research in those areas and on *Acinetobacter* species recovered from the Rivers.

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
