# Peer review of "Antibiogram Profile of Acinetobacterbaumannii Recovered from Selected Freshwater Resources in the Eastern Cape Province, South Africa"

_pathogens, 2021, doi:10.3390/pathogens10091110_

Round 1

Reviewer 1 Report

The manuscript entitled “Antibiogram profile of Acinetobacter baumannii recovered from selected freshwater resources in the Eastern Cape Province, South Africa” aims to characterize the impact of wastewater efflux, farm runoff, and hospital efflux on A. baumannii antibiotic resistance.  While this research is very important, it is difficult to draw conclusions about these environmental impacts on antibiotic resistance without more information such as where these effluxes are in relation to sample collection sites, the content of antibiotics in this water sources, etc.  There are several issues that should be addressed as follows:

  1. Line 10: Acinetobacter isolates have been found in a variety of environments including the soil.
  2. The sentence “PAR of the confirmed isolates was assessed 15 using a panel of 11 antibiotics by disc diffusion method, and ARGs in isolates exhibiting PAR; Resistance to antibiotic followed: piperacillin-tazobactam (11.2%), ceftazidime (12%), cefotaxime (18.8%), cefepime (8.8%), imipenem (2.7%), and meropenem (4.15%), amikacin (2.4%), gentamicin (8.8%), tetracycline (16.8%), ciprofloxacin (11%) and trimethoprim/sulfamethoxazole (20.5%).” Needs reworded for clarity.
  3. Were the livestock from nearby farms given antibiotics? Which antibiotics? Were the water sources tested for the presence of residual antibiotics?
  4. Line 42: Would suggest also adding this species demonstrated genetic plasticity. 
  5. Lines 56-62: This paragraph should be merged into the previous paragraph.
  6. Several minor typographical, capitalization, italic, grammar and syntax errors throughout that should be addressed such as: line 21: Imipenem and meropenem should not be capitalized; line 30-31: baumannii is misspelled; line 48: Should read “antibiotic” not “antibiotics.”
  7. Line 198: While a p value of 0.0403 is significant, it is not “highly” significant.
  8. Labeling farms, areas of wastewater efflux and hospitals on the maps would be helpful especially around areas with isolates displaying significant drug resistance. Are there more of these types of sources on the Keiskamma River?
  9. Concentrations of components of PCR reactions should be stated to allow for repetition.

Author Response

Response to reviewer 1

Point 1: Line 10: Acinetobacter isolates have been found in a variety of environments including the soil.

Response: Other natural environment such as soil, food, plant, animals’ e.t.c. have been included in the revised manuscript as suggested.

Point 2: The sentence “PAR of the confirmed isolates was assessed 15 using a panel of 11 antibiotics by disc diffusion method, and ARGs in isolates exhibiting PAR; Resistance to antibiotic followed: piperacillin-tazobactam (11.2%), ceftazidime (12%), cefotaxime (18.8%), cefepime (8.8%), imipenem (2.7%), and meropenem (4.15%), amikacin (2.4%), gentamicin (8.8%), tetracycline (16.8%), ciprofloxacin (11%) and trimethoprim/sulfamethoxazole (20.5%).” Needs reworded for clarity.

Response: The sentence has been rephrased in the revised manuscript as suggested.

Point 3: Were the livestock from nearby farms given antibiotics? Which antibiotics? Were the water sources tested for the presence of residual antibiotics?

Response: This study mainly focuses on water samples and investigated for the presence antibiotic resistant Acinetobacter baumannii, which ideally are not supposed to be found in a freshwater environment. We didn’t investigate whether the livestock in nearby farms were given antibiotics or not, but it is a normal practice to give drug to livestock. Testing for residual antibiotics in the water sources was not part of the focus of this study but may be proposed or considered in our future research.

Point 4: Line 42: Would suggest also adding this species demonstrated genetic plasticity. 

Response: This statement has been included in the revised manuscript.

Point 5: Lines 56-62: This paragraph should be merged into the previous paragraph.

Response: The paragraph has been merged with the previous as suggested.

Point 6: Several minor typographical, capitalization, italic, grammar and syntax errors throughout that should be addressed such as: line 21: Imipenem and meropenem should not be capitalized; line 30-31: baumannii is misspelled; line 48: Should read “antibiotic” not “antibiotics.”

Response: All errors highlighted here have been corrected in the revised manuscript.

Point 7: Line 198: While a p value of 0.0403 is significant, it is not “highly” significant.

Response: This statement has been corrected as suggested.

Point 8: Labeling farms, areas of wastewater efflux and hospitals on the maps would be helpful especially around areas with isolates displaying significant drug resistance. Are there more of these types of sources on the Keiskamma River?

Response: Keiskamma River has exposure to different anthropogenic activities just like other rivers but its level of significance in line with antibiotic resistance was higher than others.

Point 9: Concentrations of components of PCR reactions should be stated to allow for repetition.

Response: This has been included in the revised manuscript.

Reviewer 2 Report

This study aimed to investigate the anthropogenic impacts on freshwater environments, as represented by Acinetobacter baumannii isolates in water samples of three different rivers in South Africa.

Data presented here provide interesting insights in resistance rates in freshwater sources in South Africa by sampling five locations of three different rivers. Thus, these data will add information to resistance rates of microorganisms in freshwater environments.

However, many different human impact sources were addressed, such as wastewater treatment plant effluents, animal husbandry, waste disposal, etc, which may lead to different impacts of resistance rates and thus blur potential effects. Descriptions of sampling sites do not clarify whether the distribution of these sites allow for direct comparisons between rivers. Sample locations were also not described in relation to potential human impacts or why they were chosen specifically. It is not clear whether samples were taken up- and downstream of potential anthropogenic impact sources. Without control samples, it is not possible to reveal indications for cause-effect relationships. Collecting up- and downstream samples of each human disturbance site would strengthen the potential of this study to reveal anthropogenic impacts on resistance rates. They would allow direct comparisons of control versus test samples per sampling site using paired statistical tests.

This study clearly provides valuable data on resistance rates of microorganisms in freshwater environments, but it does not allow to directly draw conclusions on human impacts on antimicrobial resistance.

Materials and Methods

Lines 277-286: Descriptions of sampling locations do not clarify whether samples were taken up- AND downstream of an area with human impact. Furthermore, there is no clear description of the actual type of human impacts at a specific sampling location (WWTPs, livestock farming, hospital facilities, etc). Please give more specific information to clarify locations of sampling sites in relation to potential human impacts (see general comments above).

Lines 314-315: Were phenotypic susceptibility tests performed on subcultures of each colony forming unit of A. baumannii detected on original agar plates?

Table 3: Cefepime is a Cephem

Lines 350-351: A short description for which analyses correlations were used and why they were used is missing.

Results

General comment concerning statistical analyses: presentation of results of statistical analyses should contain the statistical test used, sample size/degrees of freedom, test statistic and an exact p-value: an indifferent "(p<0.05)" does not add information.

General comment concerning results of molecular data: Reporting resistance genes detected, their frequency of occurrence and showing percentage of recorded phenotypic resistances explained by detected resistance genes may be more informative than presenting heat maps and calculating Pearson correlations.

General comment concerning MDR: An MDR phenotype definition is missing and should be added to the materials and methods section. Three resistances detected for antibiotic agents of the same category will be not as significant as three resistances detected in antibiotics belonging to different antimicrobial groups each. Grouping isolates in resistances detected in one or more antimicrobial groups may clarify patterns among MDR and simplify their discussion.

Line 84: Can this significant difference be explained by a wider intermediate zone compared to other antibiotic substances? Comparison of susceptibility/resistance rates might be simplified by creating a binary variable defining S as "susceptible" and I+R as "non-susceptible".

Lines 101-106: Is "resistant" defined as isolates tested "R" in phenotypical susceptibility tests or does "resistant" also include isolates tested "I" (intermediate resistance)?

Figure 1 and lines 96-107: grouping antibiotics into antibiotic classes may simplify the heatmap and analyses of MDR phenotypes: e.g. figure 1 resistance pattern 8 (counted from above) tested resistant to CPM, CTX and CAZ. This shows resistance to three antibiotics, but these antibiotics belong to the same antibiotic group (cephem), whereas resistance pattern 1 (counted from above) tested resistant to CTX, GM and TS, belonging to three different antibiotic classes.

Figure 1: Does TS refer to Trimethoprim/Sulfamethoxazol and is SXT (only used once in this figure) a synonym of TS? If this is the case it may be easier for readers to solely use abbreviations listed in table 3.

Discussion

Discussing potential reasons of differences between and within rivers would add to the value of data presented. Do incidences of resistance genes/phenotypical resistance increase with water samples taken downstream the river, as anthropogenic impacts may accumulate from spring to estuary?

Also discuss potential reasons why phenotypic resistance was not mirrored by ARG detection.

Line 208: Correlations were not significant in two of three rivers; thus, this conclusion cannot be drawn in general.

Conclusion

General comment: The conclusion that anthropogenic impacts led to the presence of resistances cannot be drawn from data presented in this study. To reveal anthropogenic impacts on the prevalence of resistance genes, paired sampling designs (taking water samples up- and downstream from human activities) as well as from control sites or rivers less affected by human impacts (if feasible) are necessary.

Line 355: An emergence of resistance genes could not be detected in this study, as rivers were not sampled repeatedly over time to detect temporal changes of phenotypical resistance or resistance genes. Thus "detection" rather than "emergence" is more appropriate.

Author Response

Response to reviewer 2

Materials and Methods

Point 1

Lines 277-286: Descriptions of sampling locations do not clarify whether samples were taken up- AND downstream of an area with human impact. Furthermore, there is no clear description of the actual type of human impacts at a specific sampling location (WWTPs, livestock farming, hospital facilities, etc). Please give more specific information to clarify locations of sampling sites in relation to potential human impacts (see general comments above).

Response: Detailed descriptions of each sampling site have been previously published (see https://www.mdpi.com/1660-4601/17/10/3606/htm). We didn’t include this so as to avoid duplication of information. 

Point 2

Lines 314-315: Were phenotypic susceptibility tests performed on subcultures of each colony forming unit of A. baumannii detected on original agar plates?

Response: No. Not all A. baumannii colonies on original agar plates were subcultured because I was processing large number of samples. However, we follow a different pattern – distinct colonies showing the characteristics of A. baumannii on the selective medium were randomly selected from each plate and then subculture before phenotypic susceptibility tests were performed.

Point 3

T able 3: Cefepime is a Cephem

Response: The typographical error has been corrected on the Table as pointed out.

Point 4

Lines 350-351: A short description for which analyses correlations were used and why they were used is missing.

Response: updated

Results

Point 5

General comment concerning statistical analyses: presentation of results of statistical analyses should contain the statistical test used, sample size/degrees of freedom, test statistic and an exact p-value: an indifferent "(p<0.05)" does not add information.

Response: Updated with the required test statistic. But I think that, since factorial ANOVA was employed for the analyses of the data in this work the test statistic might not be necessary. The ‘sample size/degrees of freedom and p-values’ look repetitive. I prefer the previous style (p<0.5) actually. This one is actually looking inconsistent!

Point 6

General comment concerning results of molecular data: Reporting resistance genes detected, their frequency of occurrence and showing percentage of recorded phenotypic resistances explained by detected resistance genes may be more informative than presenting heat maps and calculating Pearson correlations.

Response: Table 1 and 2 showed the frequency of the phenotypic resistance and the genes detected, but the Heatmap summarizes the data better. Also the correlation provides brief information on the relationship between the phenotypic resistance and the ARG in each river

Point 7

General comment concerning MDR: An MDR phenotype definition is missing and should be added to the materials and methods section. Three resistances detected for antibiotic agents of the same category will be not as significant as three resistances detected in antibiotics belonging to different antimicrobial groups each. Grouping isolates in resistances detected in one or more antimicrobial groups may clarify patterns among MDR and simplify their discussion.

Response: updated

Point 8

Line 84: Can this significant difference be explained by a wider intermediate zone compared to other antibiotic substances?

Comparison of susceptibility/resistance rates might be simplified by creating a binary variable defining S as "susceptible" and I+R as "non-susceptible".

Response: Well, I followed the CLSI guideline on this aspect. Taking up your suggestion would mean that I should reference the source and the guideline I used during the lab work, which I may not be able to justify.

 Point 9

Lines 101-106: Is "resistant" defined as isolates tested "R" in phenotypical susceptibility tests or does "resistant" also include isolates tested "I" (intermediate resistance)?

Response: The definition of “resistant” here are isolates tested “R” and it did not include those tested “I”.

Point 10

Figure 1 and lines 96-107: grouping antibiotics into antibiotic classes may simplify the heatmap and analyses of MDR phenotypes: e.g. figure 1 resistance pattern 8 (counted from above) tested resistant to CPM, CTX and CAZ. This shows resistance to three antibiotics, but these antibiotics belong to the same antibiotic group (cephem), whereas resistance pattern 1 (counted from above) tested resistant to CTX, GM and TS, belonging to three different antibiotic classes.

Response: Nice suggestion! Updated.

Point 11

Figure 1: Does TS refer to Trimethoprim/Sulfamethoxazol and is SXT (only used once in this figure) a synonym of TS? If this is the case it may be easier for readers to solely use abbreviations listed in table 3.

Response: Yes. The correction has been effected in the revised manuscript.

Discussion

Point 12

Discussing potential reasons of differences between and within rivers would add to the value of data presented. Do incidences of resistance genes/phenotypical resistance increase with water samples taken downstream the river, as anthropogenic impacts may accumulate from spring to estuary?

Response: Updated

Point 13

Also discuss potential reasons why phenotypic resistance was not mirrored by ARG detection

Response: This is stated in the conclusion as ‘…exposure to antibiotics due to anthropogenic activities, especially overuse of drug in the livestock farming in the sampling areas. However, some of the resistant strains might be using other mechanisms to mediate resistance to antimicrobial agents besides harbouring the resistance genes’.

Point 14

Line 208: Correlations were not significant in two of three rivers; thus, this conclusion cannot be drawn in general.

Response: The conclusion did not emphasize the strength of the correlation but simply mentioned that there was a positive correlation in the all the rivers.

Conclusion

Point 15

General comment: The conclusion that anthropogenic impacts led to the presence of resistances cannot be drawn from data presented in this study. To reveal anthropogenic impacts on the prevalence of resistance genes, paired sampling designs (taking water samples up- and downstream from human activities) as well as from control sites or rivers less affected by human impacts (if feasible) are necessary.

Response: Given the nature of the study i.e. surveillance, there was no assertion that human activities actually caused the resistance observed here. But we have been able to draw attention to the fact that using such water sample portend a dangerous attempt to the public health. Your suggestion on the study design will be considered further in our work in the near future – unfortunately it is not feasible to do so at the moment due to paucity of funds.

Point 16

Line 355:  An emergence of resistance genes could not be detected in this study, as rivers were not sampled repeatedly over time to detect temporal changes of phenotypical resistance or resistance genes. Thus "detection" rather than "emergence" is more appropriate.

Response: The correction has been effected in the revised manuscript.

Round 2

Reviewer 1 Report

F values do not need to be included.

P values of 1.0000 and 0.0000 are stated.  It is likely that these are in error.  Please correct this.  P values can be reported as ≤ 0.05, 0.01 or 0.001.

Line 2: "and so on" should be removed.

Controls and experiments are absent to prove a causal relationship between resistance and human influence; therefore this study is more of a survey which itself is of value.  The manuscript should be rephrased to emphasize this.  While the experiments are absent to conclude this, it would be interesting to see if watersheds with more resistant isolates have more farming sites and hospitals.  Please include this information on the maps. 

Since it is normal practice to give antibiotics to livestock, please indicate which antibiotics are typically used for this purpose in the region.

Please indicate that testing for residual antibiotics in the water sources will be part of future research.

The manuscript needs another round of proofing as some typographical, capitalization, italic, grammar and syntax errors remain.

Concentrations of components of PCR reactions should be stated to allow for repetition. This was not addressed as indicated.

Author Response

Response to Reviewer 1 (Round 2)

P values of 1.0000 and 0.0000 are stated.  It is likely that these are in error.  Please correct this.  P values can be reported as ≤ 0.05, 0.01 or 0.001.

Response: corrected. I actually have conflicting idea on this given the comment of other reviewers. I have decided to leave as you have indicated

Line 2: "and so on" should be removed.

Response: This has been corrected.

Controls and experiments are absent to prove a causal relationship between resistance and human influence; therefore this study is more of a survey which itself is of value.  The manuscript should be rephrased to emphasize this.  While the experiments are absent to conclude this, it would be interesting to see if watersheds with more resistant isolates have more farming sites and hospitals.  Please include this information on the maps. 

Response: Yes, this is more of surveillance study on the presence of A. baumanni and its ARGs in freshwater resources in the studied locations, experiments on the relationship between resistance and anthropogenic impacts were not conducted. However the information regarding inclusion of farming sites and hospitals has been addressed in the revised manuscript.

Since it is normal practice to give antibiotics to livestock, please indicate which antibiotics are typically used for this purpose in the region.

Response: updated

Please indicate that testing for residual antibiotics in the water sources will be part of future research.

Response: Updated

The manuscript needs another round of proofing as some typographical, capitalization, italic, grammar and syntax errors remain.

Response: The manuscript has been rephrased and corrections made as pointed out.

Concentrations of components of PCR reactions should be stated to allow for repetition. This was not addressed as indicated.

Response: This has been included in the revised manuscript.

Reviewer 2 Report

The manuscript has clearly improved. There are still a few issues I would like to address or clarify:

Concerning point 1

It is true that detailed descriptions of the study sites can be found elsewhere. However, for more clarity and simplification for readers, it is still important to shortly summarise the most important characteristics of the study sites concerning potential anthropogenic effects and differences between sites. For example, pointing out potential factors (if there are) leading to higher prevalence of MDR strains and specific resistance genes in river KE, and specifically at sampling site KE2.

Concerning point 2

How did you randomly pick colonies? Did you pick a fixed number or percentage of colonies per sample (e.g. 10 colonies/sample site or every 10th colony showing characteristics of A. baumannii/sample site)? This information should be added in the materials and methods section.

Concerning point 4

The description and rationale why Pearson correlations were used should be added to the materials and methods section.

The argument that correlation analyses were used to indicate possible anthropogenic effects is not entirely convincing, as unknown genes, porin loss and upregulations of efflux pumps can also lead to resistance and could also be triggered by human interactions, but they are not detectable by identification of specific genes.

Description of any post-hoc tests also need to be added to the materials and methods section.

Concerning point 5

Test statistic and sample sizes must be stated that readers can assess the validity of the statistical analysis presented (see, for example: https://www.statology.org/how-to-report-anova-results/).

Concerning point 8

How would a significant difference in intermediate resistancesbe inerpreted?

Concerning point 9, 10 and 11

Great updates!

Concerning point 14

A correlation coefficient informs about direction and strength of a correlation. P-values however, shows whether correlations are significant, i.e. whether it is likely that there is an association between two variables (see: https://journals.lww.com/anesthesia-analgesia/fulltext/2018/05000/correlation_coefficients__appropriate_use_and.50.aspx)

Discussion, general comments

Are there distinct characteristics of KE2, GF4 and GF5 which could lead to a higher prevalence of specific resistance genes or MDR strains?

Discussion

The following statement: "The antibiogram profile of A. baumannii was evaluated in this work to understand the anthropogenic impact in the study area." Cannot be substantiated by the data presented.

However, it is possible to state, for example: "The antibiogram profiles of A. baumannii evaluated in this study revealed MDR isolates at several sites in three rivers in South Africa."

The following statement is not supported by data presented and needs rewording, since only one of three correlations was significant: "While A. baumannii in the freshwater resources were generally susceptible to antibiotics, few of the isolates exhibited MDR and the correlation between the PAR and ARG was positive in all the rivers."

The same is true for conclusions, second sentence and for the statement page 10: "In addition to the MDR phenotypes and ARGs, there was a positive correlation between PAR and ARGs in A. bau-mannii across the rivers, while Keiskamma river possessed the strongest level of correla-tion at a p-value of 0.0403 (Figure 3)."

Author Response

Response to Reviewer 2 (Round 2)

Concerning point 1

It is true that detailed descriptions of the study sites can be found elsewhere. However, for more clarity and simplification for readers, it is still important to shortly summarise the most important characteristics of the study sites concerning potential anthropogenic effects and differences between sites. For example, pointing out potential factors (if there are) leading to higher prevalence of MDR strains and specific resistance genes in river KE, and specifically at sampling site KE2.

 Response: Updated

Concerning point 2

How did you randomly pick colonies? Did you pick a fixed number or percentage of colonies per sample (e.g. 10 colonies/sample site or every 10th colony showing characteristics of A. baumannii/ sample site)? This information should be added in the materials and methods section.

Response: 20 distinct colonies of A. baumannii isolates were picked for subculturing from each sampling site for a period of one year, but this number was not consistent due to seasonal variations. For instance, the number of colonies on agar plates reduced significantly during winter season as compared to other seasons; therefore lesser number (Ë‚20) of colonies were subcultured.                                                                                                                                                                                                            

Concerning point 4

The description and rationale why Pearson correlations were used should be added to the materials and methods section.

The argument that correlation analyses were used to indicate possible anthropogenic effects is not entirely convincing, as unknown genes, porin loss and upregulations of efflux pumps can also lead to resistance and could also be triggered by human interactions, but they are not detectable by identification of specific genes.

Description of any post-hoc tests also need to be added to the materials and methods section.

 Response: ‘The argument that correlation analyses were used to indicate possible anthropogenic effects is not entirely convincing’…I guess you didn’t understand my thought on this point. I only summarise the discussion aspect. I have added your good suggestion on Pearson correlations and the description of post-hoc tests to the materials and methods section.

Concerning point 5

Test statistic and sample sizes must be stated that readers can assess the validity of the statistical analysis presented (see, for example: https://www.statology.org/how-to-report-anova-results/).

 Response: This point is conflicting with the other idea. I have decided to use p<0.05, which is generally acceptable

Concerning point 8

How would a significant difference in intermediate resistances be inerpreted?

 Response: I have collapsed and created a column for non-susceptible!

Concerning point 9, 10 and 11

Great updates!

Concerning point 14

A correlation coefficient informs about direction and strength of a correlation. P-values however, shows whether correlations are significant, i.e. whether it is likely that there is an association between two variables (see: https://journals.lww.com/anesthesia-analgesia/fulltext/2018/05000/correlation_coefficients__appropriate_use_and.50.aspx)

 Response: Updated. Sorry I missed that point due to many things to handle at a time!!

Discussion, general comments

Are there distinct characteristics of KE2, GF4 and GF5 which could lead to a higher prevalence of specific resistance genes or MDR strains?

Response: Updated

Discussion

The following statement: "The antibiogram profile of A. baumannii was evaluated in this work to understand the anthropogenic impact in the study area." Cannot be substantiated by the data presented.

However, it is possible to state, for example: "The antibiogram profiles of A. baumannii evaluated in this study revealed MDR isolates at several sites in three rivers in South Africa."

 Response: Reworded

The following statement is not supported by data presented and needs rewording, since only one of three correlations was significant: "While A. baumannii in the freshwater resources were generally susceptible to antibiotics, few of the isolates exhibited MDR and the correlation between the PAR and ARG was positive in all the rivers."

The same is true for conclusions, second sentence and for the statement page 10: "In addition to the MDR phenotypes and ARGs, there was a positive correlation between PAR and ARGs in A. bau-mannii across the rivers, while Keiskamma river possessed the strongest level of correla-tion at a p-value of 0.0403 (Figure 3)."

Response: Well, to the best of my knowledge, there is difference between positive Pearson correlations and significant Pearson correlation. The data clearly showed that there is positive correlation. For the fact we have resistance and antibiotic resistance genes occurring together in those places indicated that there is a positive correlation though all resistances might not be caused by the ARGs. Also the first heatmap showed that KE had a lot of resistance particularly at KE2 which makes the coefficient of correlation at Keiskamma significant.

Round 3

Reviewer 1 Report

All requested revisions have been addressed.  I would suggest the following minor changes:

Do not use trade names such as tenaline.  Use generic names.

Add concentrations for primers and template used for PCR.

Use "I" for intermediate resistance throughout.  The are still some instances of intermediate resistance being represented with "IR."

Author Response

Reviewer

Do not use trade names such as tenaline.  Use generic names.

Response: corrected - changed to tetracycline

Add concentrations for primers and template used for PCR.

Response: the primer concentration has been added. For DNA template, we could not quantify due to the lack of qPCR or nanodrop machine in our lab at that time.

Use "I" for intermediate resistance throughout.  They are still some instances of intermediate resistance being represented with "IR."

Response: corrected – changed to I